# Study on the Weldability of Copper—304L Stainless Steel Dissimilar Joint Performed by Robotic Gas Tungsten Arc Welding

**DOI:** 10.3390/ma15165535

**Published:** 2022-08-11

**Authors:** Andrei Mitru, Augustin Semenescu, George Simion, Elena Scutelnicu, Ionelia Voiculescu

**Affiliations:** 1Faculty of Materials Science and Engineering, Politehnica University of Bucharest, 313 Splaiul Independentei St., 060042 Bucharest, Romania; 2National Research and Development Institute for Gas Turbines COMOTI, 220 D Iuliu Maniu Bvd., 061126 Bucharest, Romania; 3Academy of Romanian Scientists, 3 Ilfov, 050044 Bucharest, Romania; 4Faculty of Engineering, “Dunarea de Jos” University of Galati, 800008 Galati, Romania; 5Faculty of Industrial Engineering and Robotics, Politehnica University of Bucharest, 313 Splaiul Independentei St., 060042 Bucharest, Romania

**Keywords:** robotic Gas Tungsten Arc Welding, dissimilar joint, SEM microstructure, EDAX chemical composition, microhardness

## Abstract

The welding process of dissimilar metals, with distinct chemical, physical, thermal, and structural properties, needs to be studied and treated with special attention. The main objectives of this research were to investigate the weldability of the dissimilar joint made between the 99.95% Cu pipe and the 304L stainless steel plate by robotic Gas Tungsten Arc Welding (GTAW), without filler metal and without preheating of materials, and to find the optimum welding regime. Based on repeated adjustments of the main process parameters—welding speed, oscillation frequency, pulse frequency, main welding current, pulse current, and decrease time of welding current at the process end—it was determined the optimum process and, further, it was possible to carry out joints free of cracks and porosity, with full penetration, proper compactness, and sealing properties, that ensure safety in operating conditions. The microstructure analysis revealed the fusion zone as a multi-element alloy with preponderant participation of Cu that has resulted from mixing the non-ferrous elements and iron. Globular Cu- or Fe-rich compounds were developed during welding, being detected by Scanning Electron Microscope (SEM). Moreover, the Energy Dispersive X-ray Analysis (EDAX) recorded the existence of a narrow double mixing zone formed at the interface between the fusion zone and the 304L stainless steel that contains about 66 wt.% Fe, 18 wt.% Cr, 8 wt.% Cu, and 4 wt.% Ni. Due to the formation of Fe-, Cr-, and Ni-rich compounds, a hardness increase up to 127 HV_0.2_ was noticed in the fusion zone, in comparison with the copper material, where the average measured microhardness was 82 HV_0.2_. The optimization of the robotic welding regime was carried out sequentially, by adjusting the parameters values, and, further, by analyzing the effects of welding on the geometry and on the appearance of the weld bead. Finally, employing the optimum welding regime—14 cm/min welding speed, 125 A main current, 100 A pulse current, 2.84 Hz oscillation frequency, and 5 Hz pulse frequency—appropriate dissimilar joints, without imperfections, were achieved.

## 1. Introduction

A large part of the dissimilar joints is achieved by fusion welding, or sometimes by soldering two or more materials that belong to different alloying systems, such as non-alloy steel and high alloy steel, ferrous and non-ferrous metals, or other combinations of different materials. In recent decades, the demand in the industry of multi-component structures achieved by joining materials completely different in structure and properties, such as steel and ceramics, ceramic composite materials, and polymeric materials reinforced with particles or fibers, has continuously increased [1,2,3]. The need for these type of joints is justified by economic reasons related to the manufacturing costs, weight of manufactured products, or special properties of materials required in service in different chemical environments [4]. One of the most important conditions in performing a quality dissimilar joint is to adequately select the materials, including the filler metal, which must be compatible from the metallurgical point of view [5,6,7,8]. The solidified mixture of two or three melted materials is characterized by non-homogenous chemical composition and an uneven distribution of internal stresses. Moreover, the typical structure of the fusion zone is characterized by a columnar dendrite structure, with the aspect of cast structure, which is in contact with the heat affected zone (HAZ), where the initial grains and the structure of materials were affected by the welding process. As the researchers reported in [9,10,11], the fusion zone is a mixing zone with specific chemical composition and mechanical properties, sometimes much different in comparison with those of the base and filler materials.

The main condition for obtaining a suitable dissimilar joint is the use of materials with reciprocal metallurgical compatibility. When the base materials have different chemical compositions, structures, and properties, a transition zone is formed in the HAZ, and inter-metallic compounds can be developed by a combination of elements that belong to the materials used to achieve the dissimilar welded joint [12,13]. The effects of the intermetallic compounds’ formation are the increase of the sensitivity to cracking and of susceptibility to corrosion, as well as the reduction of the ductility. Therefore, to limit the negative effects generated by the welding process, it is important to find the optimum technology that minimizes the width of the transition zone between the materials [14,15]. The susceptibility to the corrosion in HAZ of the dissimilar joint is caused by the electrochemical potential difference. According to the literature [16], this phenomenon may be a serious issue if the difference between the electrochemical potentials is high. On the other hand, the different dilation coefficients of the materials can generate a high level of residual stresses in the welded joint that must be balanced out by thermal treatment or by employing inter-metallic buffer layers. Moreover, the distribution and the magnitude of the residual stresses are not equal or evenly distributed in the dissimilar joint [17]. When the difference between the materials melting temperatures is large, one base material may be melted and the other may still be in a solid state, meaning that additional heat amount is required to completely melt the materials and to achieve the welded joint. This excess thermal energy may cause the evaporation of volatile elements or the overheating of the materials, determining the increase of the grains’ size in HAZ [18,19]. Furthermore, to avoid the oxidation phenomenon of chemical elements and to prevent the ingress and the dissolution of gases from the environment in the melted metal zone, adequate protection of the welding pool is needed. Another measure to achieve an adequate dissimilar joint with suitable properties is to pay attention to the mechanical preparation of the workpiece edges and to the cleaning process of the surfaces to be welded [20].

Applying an inadequate welding regime, the functional role of the welded structure may be compromised, due to the excessive reduction of hardness or of plasticity characteristics, decrease of thermal and electrical conductivity, or reduced corrosion resistance to the environment service conditions [21,22]. The mixing ratio of two or more different materials in the welding pool may be a significant issue in ensuring a good quality of the welded joint. The dilution rate is strongly dependent on the welding method, process parameters, and on the filler metal selected for joining the base materials [23,24].

The manufacturing of heat exchangers is a common application of welding Cu to SS, but, in this case, the main issues that must be addressed are the compactness and the tightness of the dissimilar joint, which must avoid the loss of cooling agent and be safely exploited at service temperatures of 300–400 °C. Based on critical analysis of Cu and SS welding compatibility, sound dissimilar welds have been achieved by applying the brazing [3,6] or welding [25,26,27,28,29,30,31,32,33,34,35,36,37,38,39,40] methods.

One of common welding techniques applied for joining dissimilar metals is the Gas Tungsten Arc Welding (GTAW) procedure which is recommended for joining thin sections of stainless steel or non-ferrous metal components, such as Al, Mg, or Cu alloys. Comparing to other fusion welding methods, the heat input generated by the process, which has a crucial role in achieving an appropriate quality weld, can be easily controlled by the operator. However, the efficiency of the manual GTAW process is significantly slower than most welding techniques [17,25,26,27]. Other welding methods, such as laser [8,11,12,15,16,21,28,29,30], friction welding [9,10,33,34,35,36], explosion welding [37,38], or induction welding [39,40], were employed to carry out dissimilar joints, but their application is limited by the configuration, dimension, and joint type. Several studies were focused on robotic GTAW, in order to find the relationship between the robot speed, welding current, heat input, and the wire feed [41,42,43]. Cold Metal Transfer (CMT) is an innovative technology that can be applied for welding dissimilar metals, especially for thicker materials, with good results in terms of weld bead aesthetics, controlled metal deposition and low heat-input [44]. In the article [45], it was reported the dissimilar joining of OFHC copper and AISI 304 stainless steel, but the welding was performed by classical GTAW method, without filler metal, using a DC current of 80 A, and pre-heating of the joint at 200 °C. The authors designed a modified socked-weld flange with edge joints made on short-flanged edge, and a butt joint of the flange edges with different height and thickness [45]. They investigated the fusion zone and found it as a semicircular-shaped bead with heterogeneous elemental distributions of Cu, Fe, Cr, Ni, and Mn. However, to our best knowledge, detailed information on robotic GTAW of this materials combination and quantifiable data on the diffusion phenomenon of chemical elements in the entire dissimilar joint are not reported in the scientific literature, and that justifies the need of further studies.

The originality of our work consists in optimizing the robotic GTAW technology, without addition of filler material and without preheating of materials before welding, that can be employed to carry out complex heat exchangers, made up of two circular plates of 304L stainless steel (SS), parallelly arranged, and a bundle of 288 pipes of Cu welded in opposite holes pairs machined in the SS plates. Due to the small distance between the edges of copper pipes, a precise positioning and a personalized inclination of the welding torch must be set in order to properly perform each of the weld beads. The main objectives of this original research were to evaluate the weldability of the dissimilar materials, joined by robotic GTAW, as well as the chemical composition determined by the diffusion and mixing phenomena developed during welding. By optimizing the process parameters, cracks, and porosity free joints, with complete penetration, proper compactness and sealing properties, which ensure safety in operating conditions, have been achieved. The findings, in terms of weldability and diffusion phenomena developed during welding, will improve the knowledge in the field of robotic GTAW of dissimilar metals.

## 2. Materials and Methods

### 2.1. Materials

A total of 288 copper pipes, each of them having 436 mm length, 28 mm diameter, and 1 mm wall thickness, were inserted in 28.5 mm diameter holes that were equidistantly drilled in two stainless steel plates with 714 mm diameter and 20 mm thickness. The two SS plates were positioned at a distance of 392 mm, the parallelism between their surfaces being maintained during the entire process by six stiffening rods that were inserted and fixed in six pairs of holes pairs with 18.5 mm diameter, equidistantly drilled.

The pipes’ ends were circularly welded, on 1.5 mm ± 0.5 length, with the 304L SS (X2CrNi19-11 according to EN 10088-1) plates, by applying different robotic GTAW process regimes. Due to the austenitic microstructure and low carbon content (0.03 wt.% C), the 304L SS has excellent corrosion resistance and good behavior in humid environments, as well as in severe corrosion conditions. However, to prevent hot cracking and stress corrosion cracking, several restrictions, mainly related to limiting the overheating and to rapid cooling in the range of 370 to 450 °C temperatures, where hard and brittle phases may develop, must be taken into consideration [20]. Sometimes, for certain applications, a heat treatment to reduce the stress level is applied, but, usually, it is not needed an annealing treatment after welding. The chemical composition of the 304L SS is shown in Table 1. The lack of several chemical elements from the SS composition, determined by EDS analysis, is caused by the small percentage of elements that is under the detectability limit of the chemical composition analyzer. The stainless steel is characterized by 215 HB30 hardness, 180 MPa yield strength, 460–680 MPa tensile strength, and 35–45% elongation at 20 °C. According to the Schaeffler Diagram, the microstructure of 304L SS contains austenite and 13% delta ferrite, with the Cr_eq_/Ni_eq_ ratio of 1.806 (where Cr_eq_ is calculated from the weight percentage of ferrite-forming elements, and Ni_eq_ from the weight percentage of austenite-forming elements) [19]. After the rolling process, the microstructure of SS exhibits equiaxed austenite grains and annealing twins, with elongated ferrite phases at the grains’ boundaries [46].

The pipes, used in the experiments, were made of Cu 99.9, whose chemical composition is presented in Table 2. In terms of the mechanical properties, Cu 99.9 has hardness of 100 HV (as cold-worked) and 50 HV (as annealed), 210 MPa tensile strength, and 33–137.8 MPa yield strength [47]. The low content of oxygen in copper is reflected in better weldability and a limitation of the cracking risk that could occur during the solidification of the melt pool. The microstructure of pure Cu 99.9 pipes consists of alpha phase and small inclusions of Cu_2_O, and other elements. After the hot rolling process, the microstructure exhibits small grains and annealing twins, while the oxide particles are aligned as dark islands in the rolling direction.

A significant aspect, to be taken into consideration in welding copper and stainless steel, is the design of the joint type. Copper has a great thermal conductivity that determines a fast dissipation of the heat before melting. To compensate this phenomenon, a supplementary amount of thermal energy is needed, but this heat excess will cause an extended melt pool, as well as a wider HAZ, and, consequently, a higher stress and strain level after the melt pool is completely solidified. Therefore, the heat amount necessary to melt the Cu pipes ends should be carefully adjusted and controlled, in order to obtain the appropriate volume of the fusion area that should ensure adequate mixing of materials.

### 2.2. Welding Process

As Figure 1a shows, the dissimilar joints were made by employing an 8-axis robotic system, equipped with a CLOOS CST FLEX S laser sensor. The working place was large enough to allow the movements in four directions of the welding system and the rotation of the robot arm for positioning the welding torch that makes the circular joint between the Cu pipes and the SS plates. To limit the personnel access and to avoid the accidental collisions in the welding zone, an infrared barrier was attached to the robotic system. The non-fusible tungsten electrode (WT20) alloyed with 2 ÷ 4% Th, with 3.2 mm diameter was used for welding the parent materials. The protection of the melt pool against contamination with O_2_, which causes porosity and reduces the mechanical properties of dissimilar joints, was ensured by the Ar shielding gas with purity of 99.99%, whose flow rate was 10 L/min, as well as 1 s for pre-purging time and at least two seconds for purging time.

The Cu pipes were machined outside and inside with a finger milling cutter, and then the surfaces were stripped with ZnCl_2_ solution, a maximum of 30 min. before welding, on at least 5 mm length. Using an alignment device and a perforated pattern, the height of the pipes’ edges, in relation to the SS plates surface, was controlled to be 1.5 mm ± 0.5. Each Cu pipe was fixed in four GTAW spots, without the addition of filler metal (Figure 1b). The welding torch has been tilted 60° relative to the horizontal plane, with the aim to avoid the leakage of the melt pool.

To minimize the overheating effect, the interpass temperature (200 °C) was monitored with IR FLIR E 60 infrared camera, while preheating and heat treatments were not applied before and after welding, respectively. The compression stresses, developed during the solidification of the melt pool, may promote the cracking phenomenon, but if the height of pipes ends is limited to 1.5 ± 0.5 mm, the volume of melted metal is still enough to obtain a good connection between the materials.

The circular robotic welding of the Cu pipes’ ends on the 304 SS plates was made without addition of filler metal. If the edge’s height is more than 2 mm, an excessive molten metal bath that solidifies much slower and may cause many defects, such as leakage of the melt metal, craters, or insufficient melting of 304L steel plate edges, can form during welding. A height of the copper pipes’ edges lower than 1 mm does not ensure sufficient melt pool volume to achieve a proper geometry of the weld bead and that may cause imperfections, such as insufficient throat, irregular weld bead, incomplete filling in the joint area, and, subsequently, an improper tightness of the dissimilar joint.

To identify the optimum values of the welding parameters, five sample groups were carried out, using different combinations of parameters. The values of the main process parameters—welding speed, main current, pulse current, oscillation frequency, and pulse frequency—introduced as input data in the GTAW robot software, are presented in Table 3. Based on the visual inspection and NDT testing of the welded samples, the fifth variant was identified as the optimum welding regime and employed for programming the robot.

The profile of the main current versus time, recorded during the robotic GTAW, shows that the total time required to perform a weld bead was 43 s (Figure 2). It was needed 1 s to reach the maximum value of the welding current and 2 s to reduce the main current value to zero. The time needed to decrease the welding current was determined by multiple tests and studies (1–5 robotic welding. Inappropriate appearance of the weld beads and final craters were noticed when the welding current was reduced to zero in less than 2 s (Figure 3a).

The length of each weld bead was 93 mm, calculated as sum between the pipe perimeter of 88 mm and the overlapping of 5 mm that was made by depositing molten metal over the starting length of the welding pass, with the aim to eliminate the cracking risk that may occur at the end of the process.

The output parameters, that are strongly dependent on the welding parameters (input data), are the geometry of the weld area, microhardness profile, and microstructure. In the cross section of the dissimilar joint (Figure 3b), where the 304L SS is the material (1) and the Cu is the material (2), the main dimensions of the fusion zone (FL) are the height (h) and the width (t). Based on the analysis of the experimental and quality inspection results, including the tightness testing, it was observed that a good behavior in the operational conditions is achieved when h is in the range of 0.80–2 mm, and t is 3.5 to 4.5 mm. For the joint shown in Figure 3b, the dimensions measured in the left side are h = 1.11 mm and t = 4.1 mm, and in the right side, h = 1.30 mm and t = 3.79 mm, the values being in the optimum ranges determined by experiments.

### 2.3. Testing Methods

The quality of the welded joints was inspected by non-destructive testing (NDT), starting with a visual examination, and followed by liquid penetrant testing (LPT), in accordance with SR EN 571-1:1999. Several defects, such as final craters in weld zone, insufficient throat, and uneven geometry were noticed during inspection. Based on the examination report and on the correlated analysis between process parameters and joints quality, the welding regime was optimized and dissimilar joints, free of defects, and imperfections, were achieved. The integrity of dissimilar joints was controlled by tightness testing with high pressure water (12.5 bars), for 15 min, no water loss being observed.

The metallographic samples were mechanically prepared in the LAMET Laboratory from the Politehnica University of Bucharest, by using the Buehler IsoMet^®^ 4000 precision automatic cutting machine (BUEHLER USA, Lake Bluff, IL, USA). Further, the samples’ surfaces were subjected to degreasing, grinding with 400 to 2500 grit self-adhesive abrasive papers and liquid coolant, polishing with abrasive suspension alpha alumina with 3 and 0.01 µm granulation, washing with warm water and degreasing with propanol after each grinding and polishing phase. The microstructure, the EDAX investigations and the microhardness measurement were performed in the LAMET Laboratory, employing the SEM FEI Quanta Inspect S Scanning Electron Microscope (FEI Europe B.V., Eindhoven, The Netherlands), equipped with the AMETEC EDAX Z2e chemical analyzer, and the Shimadzu HMV 2T equipment (Tokyo, Japan).

## 3. Results and Discussion

The microstructure analysis has been made in the most relevant regions of the dissimilar joint, meaning the interface between Cu and 304L SS, the diffusion zone and the HAZ developed in each parent material. The distribution maps of chemical elements reveal the mixing zones that are formed between the dissimilar materials, with the participation of chemical elements provided from each material (Figure 4a–c). Based on the EDS analysis (Figure 4d), it can be noticed that the center area of the fusion zone (W) is practically a new Cu-base alloy that contains over 77 wt.% Cu and other elements specific to the SS material (over 4 wt.% Cr, about 16 wt.% Fe and about 1.6 wt.% Ni).

An accurate chemical composition of the Cu and 304L steel mixture was determined by investigating several micro-areas. Thus, the analysis of the micro-area close to the FL shows the fusion zone as a mixing alloy with a dendritic microstructure that contains FeCr25Ni7 phase particles, dispersed in the Cu-rich matrix (Figure 5a). Along the FL, two mixing zones with different morphologies and compositions are formed, being determined by the predominant influence of the SS. The first mixing zone (MZ1), with 5–20 µm thickness, is developed in the vicinity of the FL and contains fine and elongated or partially rounded Cu-rich phases. Small, rounded, or elongated Cu-rich alloy particles, with a maximum diameter of 1 µm, are relatively uniformly dispersed in the Fe, Cr, and Ni-rich metal matrix, in which the concentration of Fe is about 60 wt.%, Cu about 20 wt.%, Cr about 16 wt.% and Ni is below 4 wt.% (Figure 5b). The second mixing zone (MZ2), close to the 304L SS, has a greater thickness (20–50 µm) and presents a coarse dendritic microstructure and larger partially globular Cu-rich phases with maximum 3 µm diameter. The EDS analysis performed on these phases showed a concentration of 85–88 wt.% Cu, 8–10 wt.% Fe, 2.25–2.8 wt.% Cr and 1.2–1.8 wt.% Ni (Figure 5c). The metallic matrix of MZ2 is a multi-element alloy, containing 66–67 wt.% Fe, 18 wt.% Cr, 8–8.6 wt.% Cu, and 4 wt.% Ni, with thin interdendritic zones, where Fe is 64–65 wt.%, Cr 21–22 wt.%, Cu 8–10 wt.%, and Ni 2.5–2.8 wt.%. This strip makes the transition to the 304L SS, ensuring the compatibility and the connection between the materials. Briefly, from the fusion zone to stainless steel, the Cu content decreases from 20 wt.% in MZ1 to around 8 wt.% in MZ2, determining formation of a great number of Cu-rich phases in the MZ1 area, and a smaller number of these phases in the MZ2 area, but with higher diameter, caused by the Cu content, as well as by the difference between the thermal conductivities of materials.

The profile of the main chemical elements, determined on a line of 200 µm length that crosses the interface between the WZ and the 304L steel, is plotted in Figure 6. As it was expected, the highest concentrations of Cu are registered in the fusion zone (left side chart) and of Fe, close to the 304L SS (right side chart). The chemical composition profile, in terms of Cu, Fe, and Cr content, indicates a high fluctuation near the fusion line, in the middle of the analyzed area, that indicates the reciprocal diffusion and formation of secondary compounds during welding. The diffusivity and the solubility of Cu in Fe were studied by researchers [48,49,50,51], at different temperatures, to estimate the hardening effect promoted by the copper precipitates. It was reported that the solubility limit of Cu in the ferromagnetic iron, determined by atom probe tomography (APT), is below 1 wt.% at 750 °C, for activation energy Q of 3.22 eV, and the maximum solubility of Cu in austenite is 2.3 wt.% at 850 °C. If copper is added in 1 wt.% in stainless steel with 20 wt.% Cr, the corrosion resistance, particularly in seawater and sulfuric acid solutions, is significantly improved, and a strengthening of the steel was noticed. The excess of Cu in the mixing zones and a short melting/solidification time can lead to the separation of the supplementary copper in precipitates with dimensions of 1–3 µm. Moreover, the presence of softer Cu precipitates, detected by SEM and EDS analysis in the transition zone between the fusion zone and stainless steel, causes a hardness decrease in the HAZ of stainless steel, as it is also demonstrated and shown in the chart of the hardness profile.

In the fusion zone, Fe separates as dendritic formations with Cr and Ni, or as Fe-rich precipitates, as it is seen in Figure 7. Chromium is another chemical element that showed important diffusion effects. As it can be observed from Figure 6, at about 50 microns distance from the fusion line, Cr diffuses rapidly and reaches a slight lower concentration, in comparison with the content in 304L steel, simultaneously participating in the alloying of the Ni, Cu, and Co embedding matrix. The distribution maps of chemical elements show the qualitative participation of the main chemical elements in the metallic matrix and phases’ formation (Figure 7). Each area represents an alloy characterized by specific chemical composition that demonstrates how the mixing and diffusion effects strongly depend on the peak temperature, forced convection of the melt pool, and on the mutual interaction between chemical elements.

A detailed analysis of the welding zone revealed the presence of two main phases: Cu-rich embedding matrix and rounded or chained dendritic formations of Fe–Cr–Ni alloy (Figure 8a). It was found that the Fe-rich globular phases contain over 62 wt.% Cu, 27 wt.% Fe, 7 wt.% Cr, and 2 wt.% Ni (Figure 8b), while in the fusion zone the chemical composition comprises over 93 wt.% Cu, over 3 wt.% Fe and Cr and Ni below the 1 wt.%.

A suggestive picture of phases developed in the fusion zone, which are determined by the mixing of chemical elements in the melt pool, is shown in Figure 9. It can be easily seen that Fe and Cr form clear separation boundaries from the Cu-rich embedding matrix and contribute together to developing the globular phase. In contrast, although it has an important role in determining the separate phase composition, Ni does not form separation boundaries with the embedding matrix, but has the tendency to diffuse into Cu.

The hardness profile (Figure 10) was built based on the measurements made with the Shimadzu HMV 2T microhardness tester (Shimadzu, Duisburg, Germany), according to ISO 6507-1:2006 and ASTM E384 standards. A load of 1.961 N was applied, as well as an indentation time of 10 s, to determine the microhardness in five indentations located in each region of the joint (Fusion zone, 304L SS, Cu, FL, HAZ of 304L SS) (Table 4).

Compared to the Cu hardness (82.7 HV_0.2_), it was found a hardness increase in weld (127 HV_0.2_), caused by the formation of phases rich in Fe, Cr, and Ni, but this value is acceptable for the Cu—304 SS dissimilar joints used in the fabrication process of heat exchangers. The progressive increase in hardness towards the 304L SS material was predictable, the profile being dependent on the distance from the fusion line, as well as on the diffusion phenomenon of chemical elements developed in the transition zone. The extent of the hardness increase is determined by the extent of the mixing zones (MZ1 and MZ2) formed in the HAZ of stainless steel.

As was expected, the microhardness values were quite different in the joint areas, this phenomenon being typical for the dissimilar welded joints. However, the process regime, appropriately designed to achieve an adequate transition between such different base materials, had a crucial role in avoiding the development of defects, such as cracks, craters, and other imperfections.

## 4. Conclusions

Qualitative dissimilar joints between copper and 304L austenitic stainless steel were achieved by GTAW robotic welding process, without filler metal. Based on the research results, several conclusions were drawn, as follows:When dissimilar materials, such as Cu and 304L SS, are welded together, the weld zone can be considered as a multi-element alloy, resulted from mixing the non-ferrous elements and iron, with preponderant participation of copper and separation of Cu- or Fe-rich globular compounds during welding. In the transition zone, developed between the fusion zone and the 304L stainless steel, a narrow double mixing zone, containing minimum of 66 wt.% Fe, 18 wt.% Cr, 8 wt.% Cu, and 4 wt.% Ni, was observed by microstructural analysis;Due to the formation of Fe-, Cr-, and Ni-rich compounds, an increase in hardness (up to 127 HV_0.2_) was noticed in the fusion zone in comparison with the hardness measured in the Cu metal (82 HV_0.2_);The optimization of the robotic welding regime was carried out by adjusting the parameters values, and, further, by analyzing the effects of the welding process on the geometry and the appearance of the weld bead. Finally, through successive adjustments of parameters, the optimum welding regime was determined (welding speed of 14 cm/min, main current 125 A, pulse current 100 A, oscillation frequency 2.84 Hz, and pulse frequency 5 Hz) and appropriate dissimilar joints, without imperfections and good results at the tightness test, were achieved.

## Figures and Tables

**Figure 1 materials-15-05535-f001:**
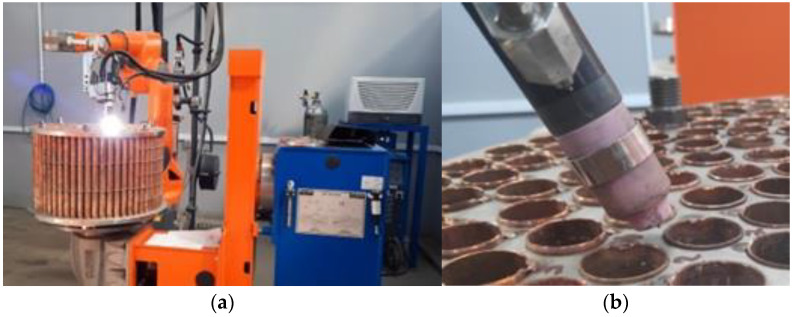
Experimental stand: (**a**) robotic welding of heat exchanger; (**b**) welding torch positioning.

**Figure 2 materials-15-05535-f002:**
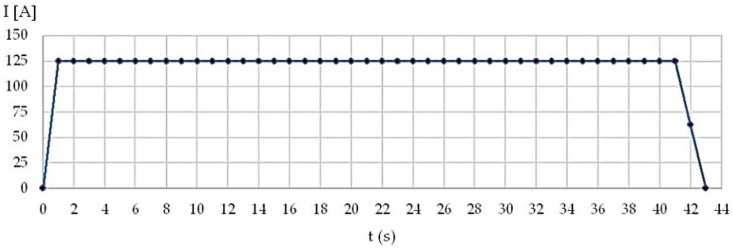
Main welding current versus time during the robotic GTAW process.

**Figure 3 materials-15-05535-f003:**
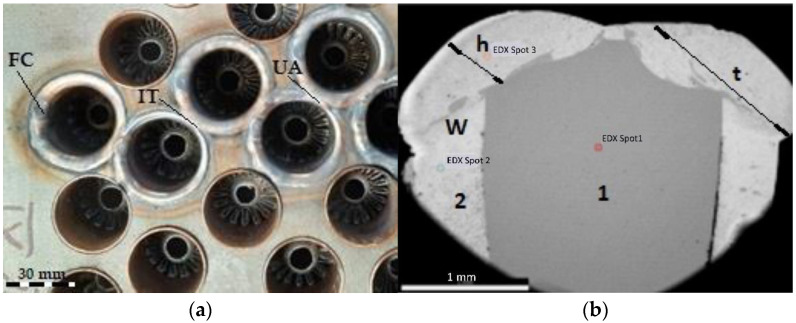
Dissimilar joints made by robotic GTAW: (**a**) imperfections of the fusion zone (FC—final crater in the fusion zone; IT–insufficient overlap; UA—uneven geometry of the weld bead); (**b**) cross-section of an adequate weld bead (1—304L SS plate, 2—Cu 99.9 pipe wall, W—fusion zone, h—fusion zone height, t—fusion zone width).

**Figure 4 materials-15-05535-f004:**
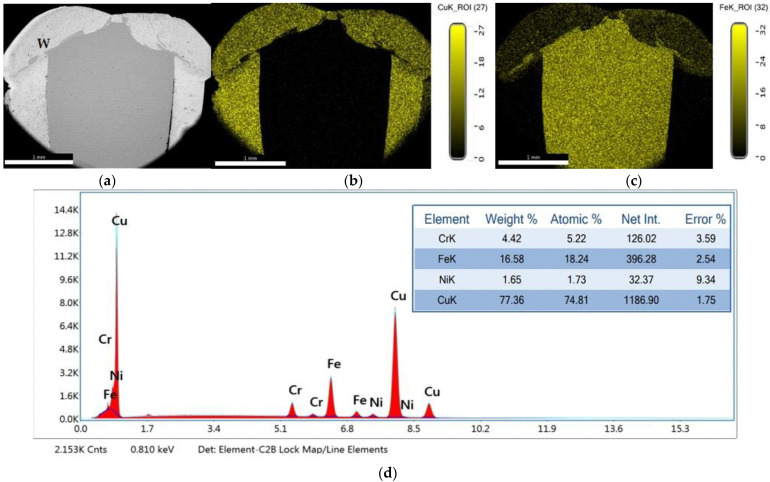
EDAX analysis and distribution maps of chemical elements: (**a**) weld macrostructure; (**b**) distribution map of Cu (**c**) distribution map of Fe; (**d**) EDS spectrum and chemical elements content in the fusion zone center.

**Figure 5 materials-15-05535-f005:**
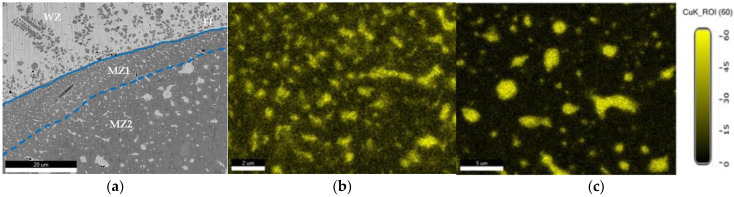
Image of the transition zone between the weld and 304L SS: (**a**) WZ—welding zone; FL—fusion line; MZ1—mixing zone 1 and MZ2—mixing zone 2; (**b**) distribution map of Cu in MZ1; (**c**) distribution map of Cu in MZ2.

**Figure 6 materials-15-05535-f006:**
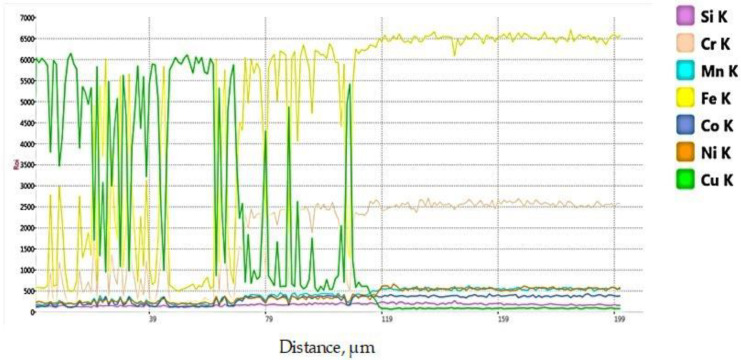
Chemical elements profile on 200 µm line that crosses the fusion line.

**Figure 7 materials-15-05535-f007:**
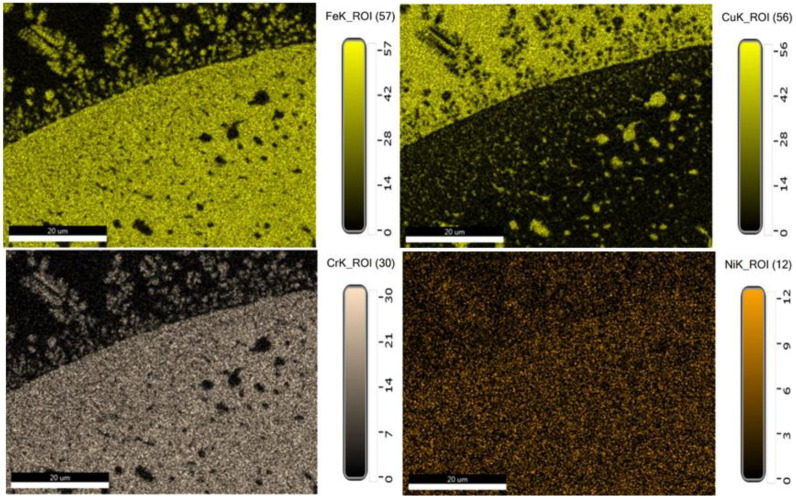
Distribution maps in HAZ for the main alloying elements: Fe, Cu, Cr, and Ni.

**Figure 8 materials-15-05535-f008:**
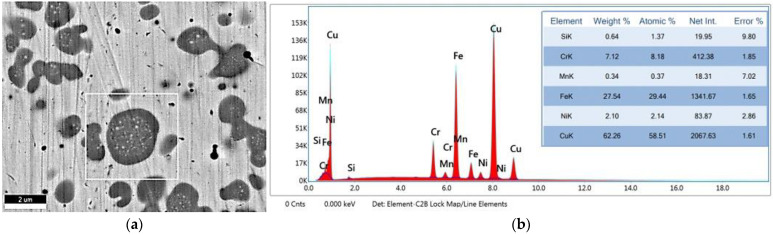
Detailed micro-zone in fusion zone: (**a**) SEM image of the micro-zone with Fe-rich phase; (**b**) EDS spectrum of the main chemical elements identified in the selected micro-zone.

**Figure 9 materials-15-05535-f009:**
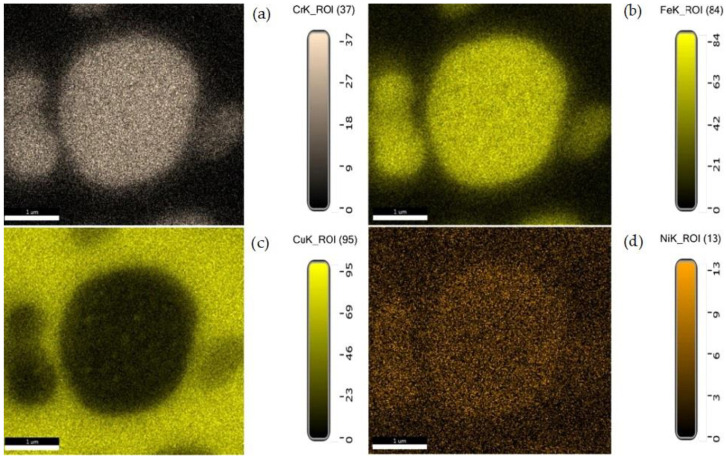
Distribution maps of the main chemical elements (Cr, Fe, Cu, and Ni) identified in the fusion zone (selected micro-zone from Figure 8: (**a**) Cr; (**b**) Fe; (**c**) Cu; (**d**) Ni).

**Figure 10 materials-15-05535-f010:**
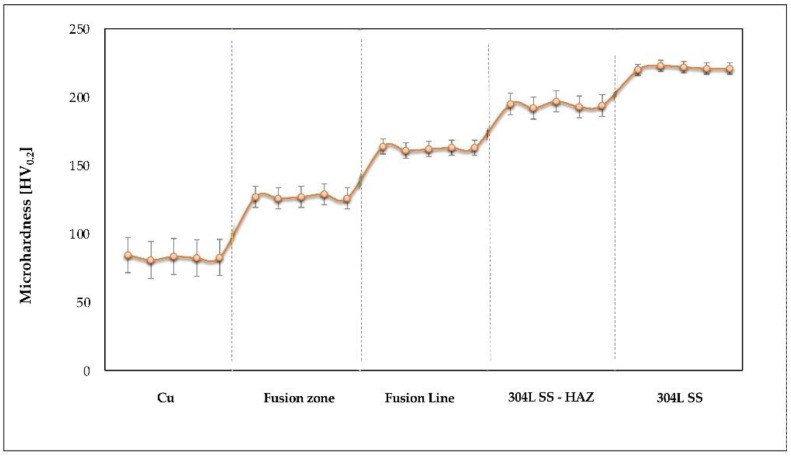
Hardness profile in the regions of the dissimilar joint.

**Table 1 materials-15-05535-t001:** Chemical composition of 304L stainless steel, in wt.%.

Method	C	Si	Mn	Cr	Ni	N	P	S	Fe
EN 10088-1	≤0.03	≤1.0	≤2.0	18.0 ÷ 20.0	10.0 ÷ 12.0	≤0.11	≤0.045	≤0.015	bal.
EDS analysis	-	0.91	1.34	19.81	9.88	-	-	-	bal.

**Table 2 materials-15-05535-t002:** Chemical composition of Cu 99.9, in wt.%, according to the supplier’s certificate.

Cu	Bi	Pb	O	Other Elements
99.9	max. 0.0005	max. 0.005	max. 0.040	0.03

**Table 3 materials-15-05535-t003:** Robotic GTAW parameters applied for performing Cu—304L SS dissimilar joints by robotic GTAW.

Variant	s_w_ Welding Speed, cm/min	f_osc_ Oscillation Frequency, Hz	f_pulse_ Pulse Frequency, Hz	I_m_ Main Current, A	I_p_ Pulse Current, A	t_d_ Decrease Time of I_m_,s	Effects of GTAW Process on Dissimilar Joint
1	12	0.25	0	140	100	1	- metal leakage - large width of weld bead - insufficient overlap
2	12	0.25	0	130	100	1	- irregular weld bead - smaller width of weld bead - deep solidification craters
3	18	0.25	0	130	100	1	- small width of weld bead - improper geometry of weld bead
4	15	0.25	0	125	100	1	- uniform width of weld bead - insufficient overlap
5	14	2.84	5	125	100	2	- optimum geometry of weld bead - imperfections-free joints

**Table 4 materials-15-05535-t004:** HV_0.2_ microhardness values measured in the Cu—304L SS dissimilar joint.

Measurement Zone		Individual Values	Average Value	Standard Deviation	Variation Coefficient
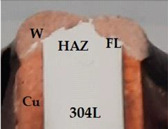	Fusion zone	127, 126, 127, 129, 126	127	1.22	0.96
	304L	220, 223, 222, 221, 221	221	1.14	0.51
	Cu	84.5, 80.8, 83.4, 82.3, 82.7	82.7	1.37	1.65
	Fusion line	164, 161, 162, 163, 163	163	1.14	0.70
	304L SS—HAZ	195, 192, 197, 193, 194	194	1.92	0.99

## Data Availability

Not applicable.

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
