# Peer review of "Study on the Weldability of Copper—304L Stainless Steel Dissimilar Joint Performed by Robotic Gas Tungsten Arc Welding"

_materials, 2022, doi:10.3390/ma15165535_

Round 1
Author Response
We are grateful for the time and effort spent to evaluate the manuscript no. materials-1848405 and we deeply appreciate the pertinent suggestions and recommendations which have led to the considerable improvement of the scientific content of the paper. Based on the reviewers’ comments, new information, written with red font, has been added in the revised manuscript. The responses are presented in the attachment.

Reviewer 2 Report
By optimizing the process parameters, sound dissimilar joints, free of defects, compact and sealed, with good adhesion between the solidified and the metals maintained in solid state during the entire welding process, have been obtained. And the assessment of weldability was only carried out to the dissimilar joint obtained by the optimized welding process parameters. It is better to described the optimizing process. The effect of process parameters on the weldability should be illustrated in the manuscript.
Author Response

(The authors gave the same response as above.)

Reviewer 3 Report
Dear authors,
This paper aims to investigate the microstructure of the weld joint with Cu and 304L SS, I will give my comments as follows:
(1) In the abstract, line 20 "a good quality joint" is mentioned, therefore properties analysis should be added, rather than the phrase in the title only "metallurgical characterization.."
(2) The introduction part is too long.
(3) Line 108- 116: The advvantage of GTAW is introduced by a comparision with laser, explosion or friction welding. However, other welding methods with filler metals, for example, cold metal transfer CMT is more effeciently used in dissimilar metal weld process.
(4) Line 123-127: As mentioned in your introduction, "detailed information on the weldability,....., made by robotic GTAW, is not reported in the scientific literature and that justifies the need of supplementary studies". I can not agree with this, because you can find many detailed research about this topic, for example, one paper was published 10 yeasr ago: Gas tungsten arc welding of CP-copper to 304 stainless steel using different filler materials https://doi.org/10.1016/S1003-6326(11)61553-7.
(5) Line 298-299: in the MZ2 area the number of Cu-rich phases is smaller but their diameter is higher compared with those in the MZ1 area, this is well shown in Fig.5, but the reason to cause such difference is not explained.
(6)The hardness increment is well observed, unfortunately, the reason of such increment is not clarified.
Author Response

(The authors gave the same response as above.)

Round 2
Author Response
Dear Reviewer,
We are grateful for the time and effort spent to revise the manuscript no. materials-1848405 and we deeply appreciate the pertinent suggestions and recommendations which have led to the considerable improvement of the scientific content of the paper. Based on your comments, information/modification, written with blue font, has been added/made in the revised manuscript. The text marked in red colour represents the modifications from the previous revision of the manuscript that have been already accepted by the reviewers. The responses are presented, in detail, in the attachment.

Reviewer 3 Report
Dear authors,
The paper is well revised. I would like to recommend it to be published.
Best regards,
Author Response
Dear Reviewer,
Thank you for your appreciation. Please, see the attachment.
